

# Significance of the inflammatory-immune-nutritional (IINS) score on postoperative survival and recurrence in breast cancer patients: a retrospective study

Yuan Wang[1], Wenxin Gao[2], Shan Wang[1], Jiajia Zhang[1], Jiaru Zhuang[1], Yibo Wu[1], Xiaoyan Huang[2] and Jing He[3]

[1] Human Reproductive and Genetic Center, Affiliated Hospital of Jiangnan University, Wuxi, China
[2] Department of Histology and Embryology, School of Basic Medical Sciences, Nanjing Medical University, Nanjing, China
[3] Department of Breast Surgery, Affiliated Hospital of Jiangnan University, Wuxi, China

Corresponding authors
Xiaoyan Huang,
bbhxy@njmu.edu.cn
Jing He, hjthy1012@163.com

## ABSTRACT

**Purpose:** Inflammation, immune system and nutritional status contribute significantly to tumorigenesis, progression and metastasis. The aim of this study was to evaluate the significance of the inflammation-immune-nutritional score (IINS) on postoperative survival and recurrence in breast cancer patients and to analyze and compare the IINS, platelet-to-lymphocyte ratio (PLR), and the prognostic nutritional index (PNI) in terms of progression-free survival (PFS) and overall survival (OS) in patients with breast cancer (BC) who underwent surgical treatment prognostic value.
**Methods:** We executed a retrospective investigation of the clinical information and related materials of 200 female breast cancer patients who had their first breast cancer operation at the Affiliated Hospital of Jiangnan University between January 2017 and December 2018, and the IINS was built using the sum of preoperative categorical scores for high-sensitivity C-reactive protein (hs-CRP), lymphocytes (LYM), and albumin (ALB). In our survival analysis, we graphed the survival curves employing the Kaplan-Meier method. The effectiveness of pre-operative IINS, PLR, and PNI in PFS and OS of breast cancer patients were evaluated with receiver operating characteristic (ROC) curves and Cox proportional risk regression analyses.
**Results:** The median age of the patients was 55.5 years (range 34–75 years). In progression-free survival, the areas under the IINS, PLR, and PNI curves were as follows: IINS: 0.735, HR (95% CI) 0.037 [0.662–0.809], PLR: 0.724, HR (95% CI) 0.036 [0.655–0.794], PNI: 0.694, HR (95% CI) 0.038 [0.619–0.769]. In overall survival, the areas under the curves of IINS, PLR, and PNI were as follows: IINS: 0.738, HR (95% CI) 0.049 [0.642–0.834], PLR: 0.700, HR (95% CI) 0.039 [0.623–0.777], and PNI: 0.713 with HR (95% CI) 0.050 [0.615–0.811]. According to the findings, among patients with resectable breast cancer, preoperative IINS may be the most accurate indicator of both overall survival and progression-free survival.

**Conclusion:** IINS may be a dependable marker for predicting postoperative survival in patients with breast cancer, and its prognostic value may be higher than that of traditional markers.

## INTRODUCTION

Breast cancer (BC) is the most prevalent malignant neoplastic disease affecting women globally (*Zannetti, 2023*). According to information on epidemiology, the disease is most common among Chinese women (*Sung et al., 2021*). Although the outlook for those diagnosed with breast cancer is relatively favorable in comparison to other forms of cancer (such as those affecting the gastrointestinal system or the lungs), the survival rates of patients with advanced or disseminated breast cancer remain poor (*D'Eredita' et al., 2001*). At present, traditional prognostic indicators include the patient's age, the TNM stage (tumor-lymph nodes-metastases), the histological grade, the estrogen receptor (ER), the progesterone receptor (PR), and the human epidermal growth factor 2 (HER2) status, all of which are employed as prognostic indicators for breast cancer (*Xie et al., 2022*). However, survival duration differs even among patients who are diagnosed at the same stage of disease and with the same causal kind of disease (*Wang et al., 2023*). Given the absence of an individualized assessment, our study employed a combination of validated biochemical indicators to examine their influence on the future prognosis of breast cancer patients (*He et al., 2023*).

In recent times, an increasing number of potential prognostic indicators that may be employed to forecast the prognosis of breast cancer have been identified and adopted in clinical practice (*Liang et al., 2021*). Prior research has unequivocally shown that dietary status and biomarkers reflecting systemic inflammatory responses are significant predictors of cancer prognosis (*Cuk et al., 2023*). Elements of systemic inflammation, such as platelets, lymphocytes and high-sensitivity C-reactive protein (hs-CRP), and biochemical markers in the blood, such as C-reactive protein levels and albumin levels, are valuable prognostic indicators for cancer, including breast cancer (*Takamizawa et al., 2020*). Currently, the platelet-to-lymphocyte ratio (PLR) and prognostic nutritional index (PNI) have been demonstrated to be reliable predictors for breast cancer patients undergoing surgical intervention, but it remains controversial as to whether they can be independent prognostic factors (*Tiainen et al., 2021*). In addition, the inflammation-immune-nutrition (IINS) value has been associated with the prognosis of hepatocellular carcinoma and colorectal carcinoma (*Li et al., 2021*). The predictive value of IINS in breast cancer has not been supported by any studies.

In order to determine the best independent prognostic markers, we retrospectively examined the progression-free survival (PFS) and overall survival (OS) of 200 breast cancer patients in relation to preoperative IINS, PLR, and PNI.

## MATERIALS AND METHODS

### Research design

We retrospectively collected clinical and pathological information from 200 female breast cancer patients who underwent first-time surgical treatment for breast cancer at the Affiliated Hospital of Jiangnan University between January 2017 and December 2018, using a convenience sampling method. Inclusion criteria were as follows: (1) All patients were pathologically diagnosed as having primary breast cancer; (2) the presence of invasive breast cancer in all of the patients was confirmed by pathological evaluations. (3) Undergoing radical surgical resection for the first time; (4) with complete clinicopathological and laboratory information. Exclusion criteria: (1) Having a malignant tumor other than breast cancer; (2) patients with acute and chronic preoperative inflammatory and infectious diseases; (3) patients treated with radiation before surgery; This study has the approval of the medical ethics committee of the Affiliated Hospital of Jiangnan University (JNMS01202300207). As all data were anonymized, the Ethics Committee waived the requirement for informed consent. Research was conducted according to the tenets of the Declaration of Helsinki.

### Data collection

Hospital electronic medical records and clinical data were used to collect pre-surgery blood biochemistry indices and pathology information from breast cancer patients. Blood samples were taken from all of the patients in the week prior to surgery. Body mass index (BMI) was classified in three categories: ≤18.5, 18.5 to 23.9 and >23.9 (*Song et al., 2022*). Each patient's cancer stage (including tumor size, axillary lymph node positivity and TNM) was determined using the American Joint Committee on Cancer (AJCC) staging manual, eighth edition (*Lu et al., 2022*). At baseline before surgery, serum biochemical tests were performed, including laboratory tests (high-sensitivity C-reactive protein, lymphocyte count, thrombocyte count and albumin). Follow-up data were collected using outpatient review and telephone-based follow-up. The preliminary postoperative follow-up for breast cancer is scheduled approximately 2 months after the surgical procedure. At this juncture, the patient undergoes a physical examination, along with laboratory and hematological tests. Subsequent follow-ups are conducted every 3 months. In the absence of recurrence within 2 years, this process can be repeated every 6 months until a 5-year follow-up is achieved. Progression events were strictly defined according to RECIST 1.1 criteria, combining contrast-enhanced computerized tomography (CT) or magnetic resonance imaging (MRI) findings, pathologic confirmation, and clinical correlation. Imaging evaluations were performed at 3-month intervals for the first 2 years and every 6 months thereafter until the end of the 5-year follow-up. The follow-up period is defined as the interval between the initiation of treatment and the date of final confirmation of patient survival or death. The follow-up period was defined as the time interval between the initiation of treatment and the date of final confirmation of patient survival or death. Overall survival (OS) was defined as the time interval between the date of surgery and the occurrence of all-cause death or the conclusion of the follow-up period.

Progression-free survival (PFS) was defined as the time interval between the date of surgery and the first indication of disease progression or the conclusion of the follow-up period (*Zehra, Ali & Zafar, 2023*).

# DEFINITION

We used X-tile software version 3.6.1 (https://medicine.yale.edu/lab/rimm/research/software/, Yale University School of Medicine, New Haven, CT, USA) to select the optimal threshold values for high-sensitivity C-reactive protein (hs-CRP), lymphocytes (LYM), and albumin (ALB) based on the association of each metric with patient overall survival (*Yeun & Kaysen, 1998*). Three categories of hs-CRP were created based on the two critical values (score 0: ≤6 mg/L, score 1: >6 and ≤12 mg/L, and score 2: >12 mg/L); whereas LYM and ALB were categorized as follows: lym (score 0: >1.6 × $10^9$/L, score 1: >1.1 × $10^9$/L and ≤1.6 × $10^9$/L, and score 2: ≤1.1 × $10^9$/L); ALB (score 0: >38.4 g/L, score 1: >34.8 g/L and ≤38.4 g/L, score 2: ≤34.8 g/L). The IINS was then calculated by adding the scores for hs-CRP, LYM, and ALB. Given that the study's median IINS was 2, an IINS of greater than 2 was considered to belong to the high IINS group. As an example, a patient's preoperative hs-CRP, LYM, and ALB values were 14.0 mg/L, 1.5 × $10^9$/L, and 30.6 g/L. The hs-CRP, LYM, and ALB scores were then 2, 1, and 2, respectively. Then, the IINS score was 5 (high IINS) (*Li et al., 2021*).

To appraise the prognosis of IINS in comparison with other markers of prediction, we also looked at the prognostic manifestations of PLR and PNI, and the optimal thresholds for PLR and PNI that are derived from the subject's work characterization (receiver operating characteristic, ROC) plots are shown in Table S1. PLR and PNI are defined as follows: PLR = platelet/lymphocyte count, PNI = albumin + 5 × lymphocyte count (*Camp, Dolled-Filhart & Rimm, 2004*).

## Statistical analysis

Statistical analysis was undertaken utilizing SPSS 26.0 software. For continuous variables, differences between groups were assessed using the t-test or Wilcoxon signed rank test. Differences between categorical variables are determined using Fisher's exact test or Chi-square test. By predicting PFS and OS for PLR and PNI, the area under the receiver operating characteristic (ROC) curve (AUC) and the ideal threshold were obtained. Survival curves were depicted utilizing the Kaplan-Meyer technique and differences were compared utilizing the log-rank test (*Dolan et al., 2018*). For univariate and multivariate regression analyses, Cox proportional risk regression models were used. To demonstrate prognostic relationships between patients with different characteristics and the proposed score, subgroup studies were performed. A comparative ROC analysis was finally performed using SAS software to assess whether the differences in AUC values between PLR, PNI, and IINS were statistically significant. For all two-way adjustments, $p$ values < 0.05 were considered statistically significant.

**Table 1 Baseline clinicopathologic characteristics of breast cancer patients.**

| | | Disease progression | | $p^*$ | Death | | $p^*$ |
|---|---|---|---|---|---|---|---|
| | | Without N = 123 (%) | With N = 77 (%) | | No N = 157 (%) | Yes N = 43 (%) | |
| Age (yr)[a] | | 55 (50–62) | 56 (51–62.5) | 0.48 | 54 (49–60) | 64 (56–69) | <0.001 |
| BMI | | | | 0.55 | | | 0.872 |
| | <18.5 | 6 (4.9) | 1 (1.3) | | 6 (3.8) | 1 (2.3) | |
| | 18.5–23.9 | 61 (49.6) | 35 (45.5) | | 76 (48.4) | 20 (46.5) | |
| | >23.9 | 56 (45.5) | 41 (53.2) | | 75 (47.8) | 22 (51.2) | |
| Tumor size | | | | 0.16 | | | 0.053 |
| | <2 cm | 62 (50.4) | 39 (50.6) | | 81 (51.6) | 20 (46.5) | |
| | >2 cm, <5 cm | 60 (48.8) | 36 (46.8) | | 75 (47.8) | 21 (48.8) | |
| | >5 cm | 1 (0.8) | 2 (2.6) | | 1 (0.6) | 2 (4.7) | |
| TNM stage | | | | <0.001 | | | <0.001 |
| | I | 56 (45.5) | 26 (33.8) | | 74 (47.1) | 8 (18.6) | |
| | II | 52 (42.3) | 17 (22.1) | | 53 (33.8) | 16 (37.2) | |
| | III | 15 (12.2) | 34 (44.2) | | 30 (19.1) | 19 (44.2) | |
| Node positivity | | | | <0.001 | | | <0.001 |
| | 0 | 93 (75.6) | 25 (32.5) | | 103 (65.6) | 15 (34.9) | |
| | 1–3 | 23 (18.7) | 22 (28.6) | | 35 (22.3) | 10 (23.3) | |
| | 4–9 | 4 (3.3) | 18 (23.4) | | 13 (8.3) | 9 (20.9) | |
| | >10 | 3 (2.4) | 12 (15.6) | | 1 (0.6) | 9 (20.9) | |
| ER | | | | 0.76 | | | 0.584 |
| | Positive | 81 (65.9) | 49 (63.6) | | 104 (66.2) | 26 (60.5) | |
| | Negative | 42 (34.1) | 28 (36.4) | | 53 (33.8) | 17 (39.5) | |
| PR | | | | 0.93 | | | 0.172 |
| | Positive | 60 (48.8) | 38 (49.4) | | 81 (51.6) | 17 (39.5) | |
| | Negative | 63 (51.2) | 39 (50.6) | | 76 (48.4) | 26 (60.5) | |
| HER-2 | | | | 0.35 | | | 0.661 |
| | Positive | 103 (83.7) | 60 (77.9) | | 129 (82.2) | 34 (79.1) | |
| | Negative | 20 (16.3) | 17 (22.1) | | 28 (17.8) | 9 (20.9) | |
| Hs-CRP | | | | <0.001 | | | <0.001 |
| | | 2.0 (2.0–4.0) | 9.0 (6.5–15) | | 3.0 (2.0–6.0) | 14 (9–15) | |
| Lym ($10^9$/L)[a] | | | | 0.02 | | | 0.022 |
| | | 1.3 (0.9–1.7) | 1.1 (0.9–1.5) | | 1.3 (0.9–1.7) | 1.0 (0.8–1.5) | |
| ALB (G/L)[a] | | | | <0.001 | | | <0.001 |
| | | 40.9 (38.9–43.7) | 39.3 (35.5–42.1) | | 40.6 (38.6–43.5) | 36.5 (32.3–42.1) | |
| PLT | | | | <0.001 | | | <0.001 |
| | | 175 (142–223) | 256 (214–307) | | 201 (150–244.5) | 280 (172–333) | |

**Notes:**

Hs-CRP, high sensitivity C-reactive protein; Lym, lymphocyte; ALB, albumin, blood platelet; MI, body mass index; ER, estrogen receptor; PR, progesterone receptor; Her-2, human epidermal growth factor receptor 2; TNM, tumor-node-metastasis.

[*] $p$-values were calculated by the student's t-test or Wilcoxon test for continuous variables, and the chi-square test for categorical variables, respectively.

[a] Age, BMI, Tumor size and Node positivity, Hs-CRP, Lym, ALB, PLT are continuous variables, the others (TNM stage, ER, PR and HER-2) are categorical variables.

## RESULTS

### Patient characteristics

The trial involved 200 patients with breast cancer, all of whom were eventually treated with surgery. Table 1 shows demographic and clinicopathological characteristics. All patients had a median age of 55.5 years (range, 34–75 years). In accordance with AJCC classification, 82 patients (41%) were stage I, 69 (34.5%) were stage II, and 49 (24.5%) were stage III. 7 (3.5%) had low weight, 94 (47%) had normal weight and 99 (49.5%) had excess weight. Regarding tumor size, 101 (50.5%) were T1, 96 (48.0%) were T2, and 3 (1.5%) were T3. 118 (59%) of the axillary lymph nodes were negative, 1–3 (45, 22.5%), 4–9 (22%) and more than 10 (7.5%) were positive. The ER status was negative in 130 patients (65%) who had a positive ER status and in 70 patients (35%) who had a negative ER status. A total of 98 patients (49%) had a positive PR status and 102 (51%) had a negative status. For the Her-2 status, 163 (81.5%) had a positive result and 37 (18.5%) had a negative result. Overall, 77 patients (38.5%) relapsed, and 43 patients (21.5%) died. The average follow-up was 46 months.

### Relationship between inflammation-immunity-nutrition scores and clinicopathologic features

In all, 68 patients had low IINS, and 132 patients had high IINS. Clinicopathological factors associated with IINS are shown in Table 2. Patients with a high IINS had an older age ($p < 0.001$), a higher TNM stage ($p = 0.001$), a larger tumor area ($p = 0.031$) and a higher number of positive axillary lymph nodes ($p < 0.001$), higher level of PLR ($p < 0.001$), higher level of high-sensitivity C-reactive protein ($p < 0.001$) and higher platelet counts ($p < 0.001$), and patients with high IINS had lower levels of BMI ($p = 0.001$) and lower levels of albumin ($p < 0.001$), lymphocytes ($p < 0.001$), and PNI ($p < 0.001$). However, IINS was not associated with estrogen, progesterone, or human epidermal growth factor.

### Univariate and multifactorial analysis of PFS and OS in breast cancer patients

Univariate and multivariate analyses were performed on the following: IINS, PLR, PNI, age, TNM stage, tumor size, and axillary lymph node positivity. One-way Cox regression analysis revealed that IINS (95% CI 1.015 [1.006–1.023], $p = 0.001$), PLR (95% CI 1.002 [1.001–1.004], $p < 0.001$), PNI (95% CI 0.899 [0.862–0.937], $p < 0.001$) and TNM staging (95% CI 1.734 [1.297–2.319], $p < 0.001$) and axillary lymph node positivity (95% CI 1.115 [1.074–1.157], $p < 0.001$)) were associated with the progression-free survival of patients (Table 3). In addition, the following were significant: IINS (95% CI 1.017 [1.008–1.026], $p < 0.001$), PLR (95% CI 1.003 [1.001–1.004], $p = 0.001$), PNI (95% CI 0.861 [0.814–0.910], $p < 0.001$) and age (95% CI 1.121 [1.076–1.168], $p < 0.001$), Tumour size (95% CI 1.213 [1.029–1.430], $p = 0.021$) and positive axillary lymph nodes (95% CI 1.136 [1.080–1.195]; $p < 0.001$) were associated with patients' overall survival (Table 4). Multifactorial analysis showed that insulin (95% CI 1.812 [1.431–2.293], $p < 0.001$) was associated with progression-free survival in patients with breast cancer. Multifactorial analysis also showed

**Table 2 Relationship between inflammation-immunity-nutrition score and clinicopathologic features.**

| | | IINS value | | |
| --- | --- | --- | --- | --- |
| | | IINS ≤ 2<br>N = 132 (%) | IINS > 2<br>N = 68 (%) | p* |
| Age (mean ± SD) | | 54 (49–60) | 59 (53–65) | <0.001 |
| BMI | | | | 0.001 |
| | <18.5 | 5 (3.8) | 2 (2.9) | |
| | 18.5–23.9 | 63 (47.7) | 33 (48.5) | |
| | >23.9 | 64 (48.5) | 33 (48.5) | |
| TNM stage | | | | 0.001 |
| | I | 62 (47) | 20 (29.4) | |
| | II | 48 (36.4) | 21 (30.9) | |
| | III | 22 (16.7) | 27 (39.7) | |
| Tumor size | | | | 0.031 |
| | d < 2 cm | 71 (53.8) | 30 (44.1) | |
| | 2 cm < d < 5 cm | 60 (45.5) | 36 (52.9) | |
| | d > 5 cm | 1 (0.8) | 2 (2.9) | |
| Node positivity | | | | <0.001 |
| | 0 | 91 (68.9) | 27 (39.7) | |
| | 1–3 | 28 (21.2) | 17 (25) | |
| | 4–9 | 9 (6.8) | 13 (19.1) | |
| | >10 | 4 (3.0) | 11 (16.2) | |
| ER | | | | 0.707 |
| | Positive | 87 (65.9) | 43 (63.2) | |
| | Negative | 45 (34.1) | 25 (36.8) | |
| PR | | | | 0.322 |
| | Positive | 68 (51.5) | 30 (44.1) | |
| | Negative | 64 (48.5) | 38 (55.9) | |
| HER-2 | | | | 0.585 |
| | Positive | 109 (82.6) | 54 (79.4) | |
| | Negative | 23 (17.4) | 14 (20.6) | |
| PLR | | | | <0.001 |
| | | 143.5 (94.050–205.125) | 244.650 (180.950–289.825) | |
| PNI | | | | <0.001 |
| | | 48.350 (45.625–51.575) | 43.250 (39.425–45.800) | |
| hsCRP | | | | <0.001 |
| | | 2.00 (2.00–4.00) | 9.00 (8.00–15.00) | |
| Lym ($10^9$ /L)[a] | | | | <0.001 |
| | | 1.45 (1.00–1.80) | 1.000 (0.800–1.300) | |
| ALB | | | | <0.001 |
| | | 41.10 (39.400–43.575) | 37.350 (33.950–41.050) | |
| PLT | | | | <0.001 |
| | | 194.5 (147.25–244.00) | 249.50 (184.74–310.50) | |

(Continued)

| | | IINS value | | |
| | | IINS ≤ 2 N = 132 (%) | IINS > 2 N = 68 (%) | p* |
|---|---|---|---|---|
| PFS | | | | <0.001 |
| | YES | 24 (18.2) | 53 (77.9) | |
| | NO | 108 (81.8) | 15 (22.1) | |
| OS | | | | <0.001 |
| | YES | 8 (6.1) | 35 (51.5) | |
| | NO | 124 (93.9) | 33 (48.5) | |
| Treatment modalities (post-operative) | | | | 0.782 |
| | chemotherapy | 43 (32.6) | 25 (36.8) | |
| | radiotherapy | 46 (34.8) | 24 (35.3) | |
| | Other | 43 (32.6) | 19 (27.6) | |

Notes:

IINS, inflammation-immunity-nutrition score; PLR, platelet count to lymphocyte count ratio; PNI, nutrient index.

* p values were calculated by the student's t-test or Wilcoxon test for continuous variables, and the Chi-square test for categorical variables, respectively.

a Age, tumor size, BMI and node positivity are continuous variable, the others (TNM stage, ER, PR and HER-2) are categorical variables.

**Table 3 Association between prognostic factors and progression-free survival in breast cancer by univariate and multivariate cox regression analysis.**

| | Cut-off | Univariate HR (95% CI) | p | Multivariate* HR (95% CI) | p |
|---|---|---|---|---|---|
| IINS | 48.75 | 1.015 [1.006–1.023] | 0.001 | 1.812 [1.431–2.293] | <0.001 |
| PLR | 159.53 | 1.002 [1.001–1.004] | <0.001 | 1.001 [1.000–1.003] | 0.108 |
| PNI | 45.15 | 0.899 [0.862–0.937] | <0.001 | 1.050 [0.986–1.118] | 0.129 |
| Age (year) | – | 1.009 [0.982–1.038] | 0.516 | | |
| TNM stage | – | 1.734 [1.297–2.319] | <0.001 | 1.166 [0.791–1.718] | 0.437 |
| Tumor size (cm) | – | 1.153 [0.990–1.343] | 0.067 | | |
| Number of positive lymph nodes | – | 1.115 [1.074–1.157] | <0.001 | 1.037 [0.977–1.099] | 0.230 |

Notes:

HR, Hazard ratio; CI, confidence interval; IINS, inflammation-immunity-nutrition score; PLR, platelet count to lymphocyte count ratio; PNI, nutrient index.

* Multivariate cox regression models included age, TNM stage, tumor size and node positivity for mutual adjustment.

Number of positive lymph nodes, the count of lymph nodes confirmed to have metastasis by pathological examination.

that IINS (95% CI 2.552 [1.740–3.742], p < 0.001), age (95% CI 1.101 [1.056–1.148], p < 0.001), and tumour size (95% CI 1.780 [1.053–3.010], p = 0.031) were associated with overall survival in breast cancer patients (Table 4).

## Prognostic value of IINS in progression-free survival

Table S1 shows the area under the ROC curve (AUC) of PFS for IINS, PLR and PNI. Results showed that IINS (AUC: 0.735; 95% CI [0.622–0.809]) predicted PFS better than PLR (AUC: 0.724; 95% CI [0.655–0.794]) and PNI (AUC: 0.694; 95% CI [0.619–0.769]). On the basis of the ROC curve, we found that the IINS had the best area under the curve with a value of 0.735 (As shown in Fig. 1). The Kaplan-Meier survival curves showed that patients in the low IINS group had a longer progression-free survival (Fig. 2). Integrating

**Table 4 Association between prognostic factors and overall breast survival by univariate and multivariate cox regression analysis.**

|  | Cut-off | Univariate HR (95% CI) | P | Multivariate* HR (95% CI) | P |
|---|---|---|---|---|---|
| IINS | 48.65 | 1.017 [1.008–1.026] | <0.001 | 2.552 [1.740–3.742] | <0.001 |
| PLR | 163.48 | 1.003 [1.001–1.004] | 0.001 | 1.002 [0.999–1.005] | 0.179 |
| PNI | 43.55 | 0.861 [0.814–0.910] | <0.001 | 1.080 [0.984–1.185] | 0.105 |
| Age (year) | – | 1.121 [1.076–1.168] | <0.001 | 1.101 [1.056–1.148] | <0.001 |
| TNM stage | – | 2.113 [1.435–3.112] | <0.001 | 3.216 [0.865–11.965] | 0.081 |
| Tumor size (cm) | – | 1.213 [1.029–1.430] | 0.021 | 1.780 [1.053–3.010] | 0.031 |
| Number of positive lymph nodes | – | 1.136 [1.080–1.195] | <0.001 | 0.933 [0.854–1.020] | 0.129 |

Notes:
HR, Hazard ratio; CI, confidence interval; IINS, inflammation-immunity-nutrition score; PLR, platelet count to lymphocyte count ratio; PNI, nutrient index.
* Multivariate cox regression models included age, TNM stage, tumor size and node positivity for mutual adjustment.
Number of positive lymph nodes, the count of lymph nodes confirmed to have metastasis by pathological examination.

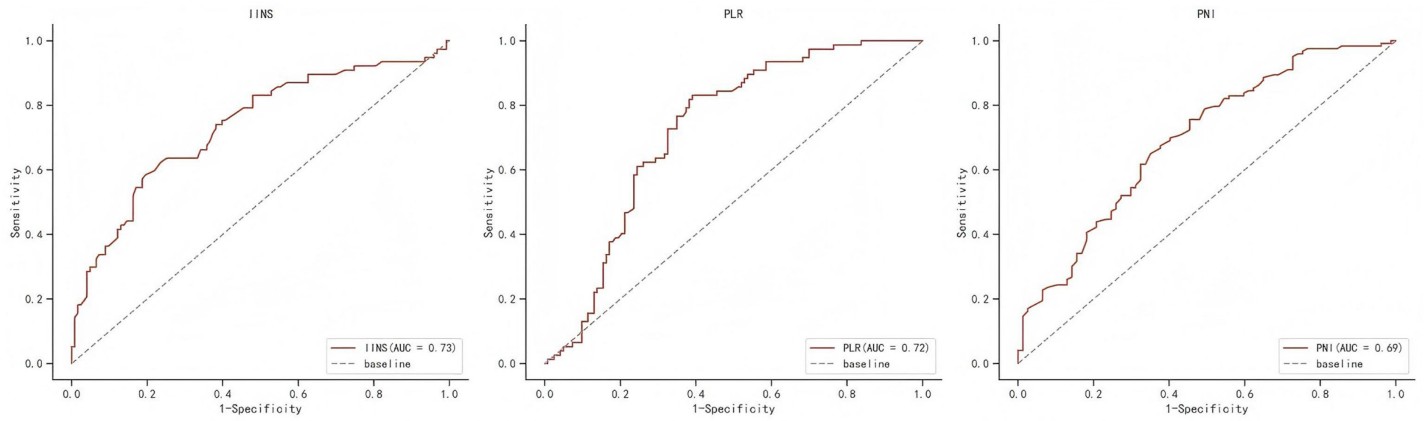

**Figure 1 The ROC curve for progression-free survival of IINS, PLR and PNI.**

the results of area under the curve and multivariate Cox regression analyses, the study confirms that IINS is a novel prognostic indicator of progression-free survival in breast cancer patients.

## Prognostic value of IINS in overall survival

Table S1 shows the area under the ROC curve (AUC) of OS for IINS, PLR and PNI. The findings revealed that IINS (AUC: 0.738; 95% CI [0.642–0.834]) was a better predictor of OS than PLR (AUC: 0.700; 95% CI [0.623–0.777]) and PNI (AUC: 0.713; 95% CI [0.615–0.811]) On the basis of the ROC curve, we found that the IINS had the best area under the curve with 0.738 points (as shown in Fig. 3). Similarly, the Kaplan-Meier survival curve showed that patients in the low IINS group had longer overall survival (Fig. 4). Integrating the area under the curve with the results of multivariate Cox regression analyses, the study confirms that IINS is also a novel prognostic indicator of overall survival in breast cancer patients.

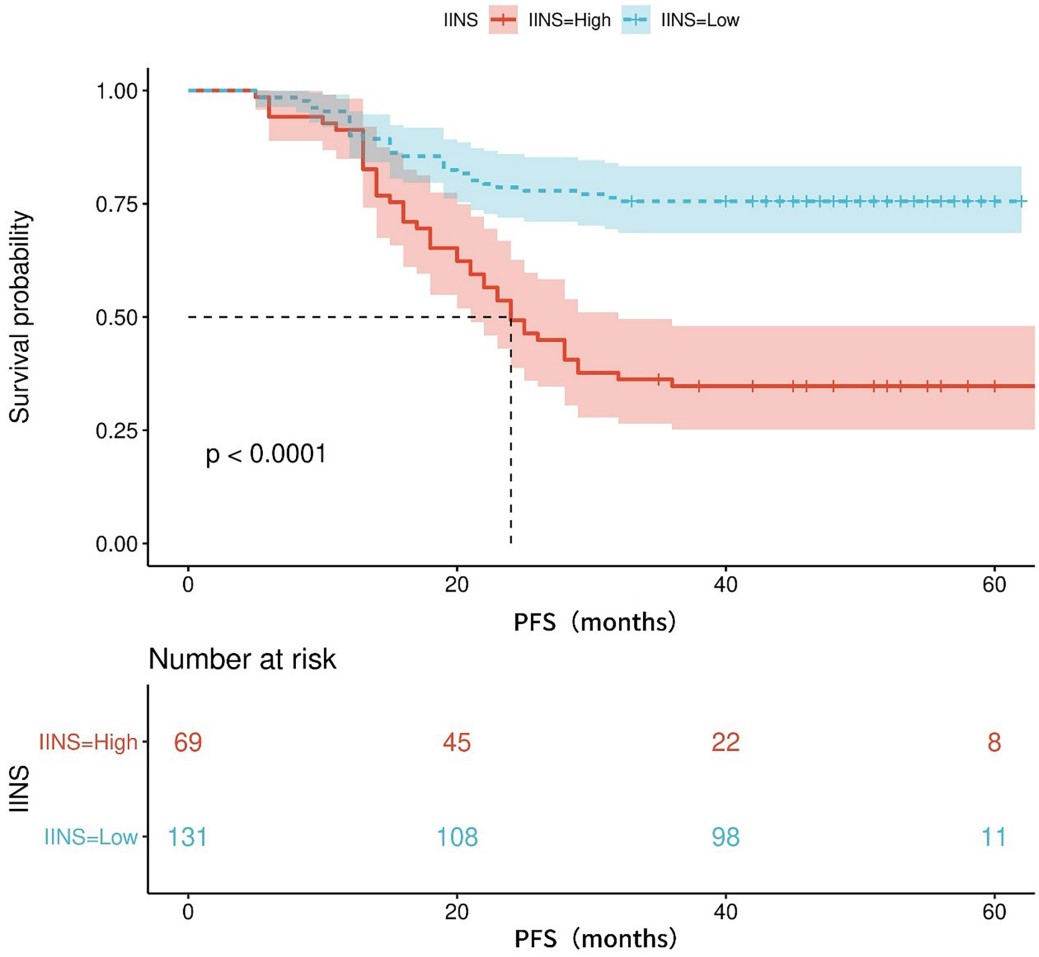

**Figure 2 The Kaplan-Meier curves for progression-free survival of breast cancer patients on IINS.**

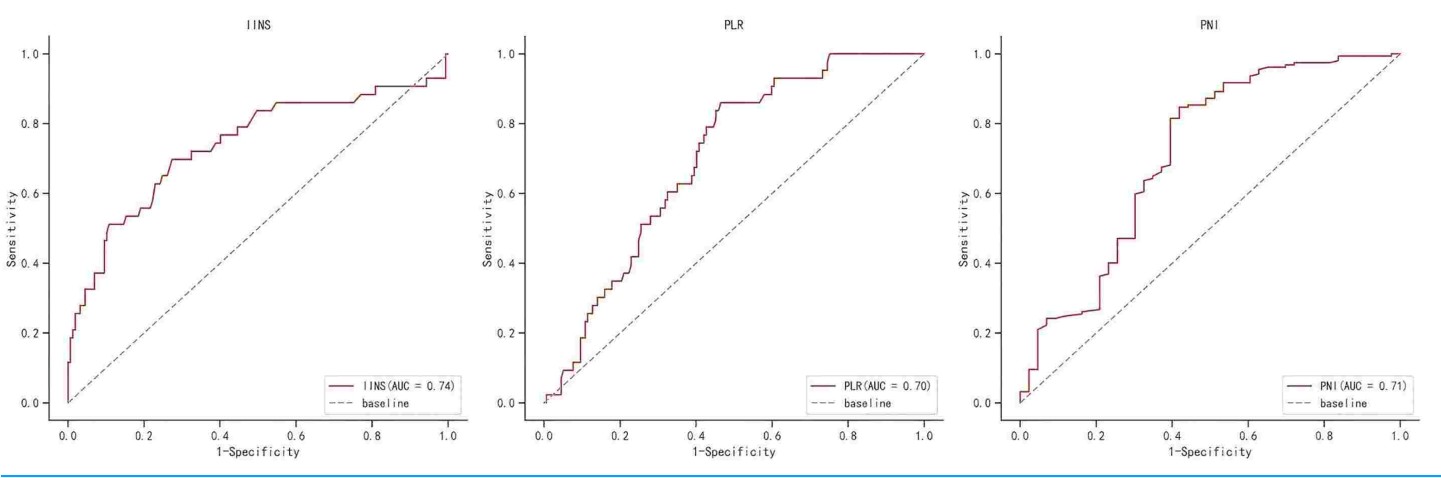

**Figure 3 The ROC curve for overall survival of IINS, PLR and PNI.**

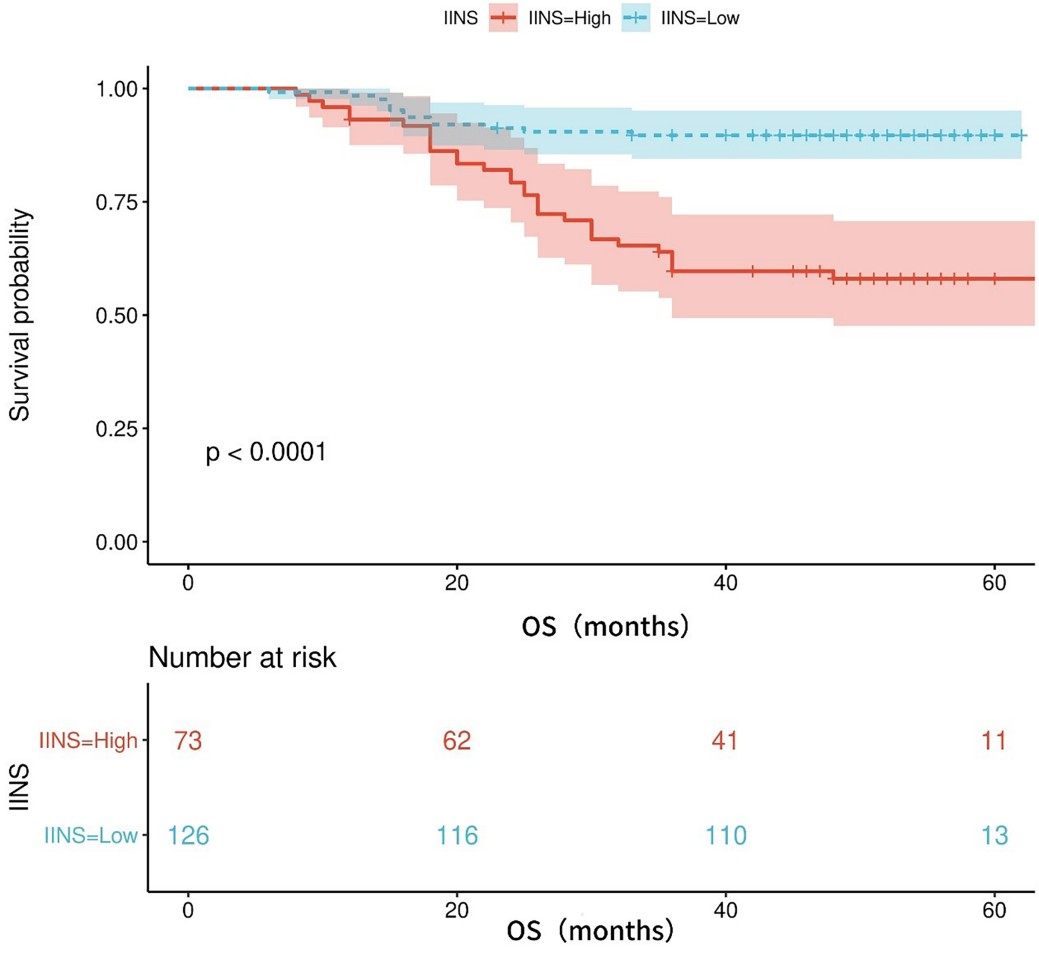

**Figure 4 The Kaplan-Meier curves for overall survival of breast cancer patients based on IINS.**

### Comparative analysis of ROC for IINS, PLR and PNI

The results of the comparative ROC analyses of IINS, PLR, and PNI are presented in Table S1. The results demonstrated that in PFS, the AUC comparisons of IINS ($p < 0.001$), PLR ($p = 0.025$), and PNI ($p = 0.046$) were all significant; in OS, the AUC comparisons of IINS ($p < 0.001$), PLR ($p = 0.034$), and PNI ($p = 0.012$) were also significant. Furthermore, the sensitivity, specificity, positive predictive value (PPV), and negative predictive value (NPV) are demonstrated in Table S2.

## DISCUSSION

In this study, we revealed that IINS might function as a reliable predictive score for those who have gone through breast cancer surgery. The findings of this study indicate that IINS is a prognostic indicator affecting BC patients, with PFS and OS being considerably better in individuals with low IINS than in patients with high IINS. Additional comparisons revealed that IINS performed better in terms of prognosis than other indices including the PLR and PNI. To the greatest extent of our knowledge, this retrospective study is the first

to appraise the predictive feasibility of the IINS in BC patients undergoing post-surgical treatment.

Blood and biochemical markers had been shown in a rising number of studies to be prognostic indicators for cancer (*Wattacheril et al., 2023*). The majority of present-day prognostic biomarkers, on the other hand, are various combinations of two markers in serum tests, which do not represent the immune and nutritional activities of the body, resulting in bias and poor prediction by default (*De Larco, Wuertz & Furcht, 2004*). A good prognostic indicator for resectable colorectal, hepatocellular, and endometrial cancers has been demonstrated to be the inflammation-immunity-nutrition score based on the preoperative hs-CRP, LYM, and ALB composite score (*Antonio et al., 2015*; *Wang et al., 2024*). The role of these indicators can also be used to explain the predictive value of IINS for breast cancer. Low immunological and nutritional status together with a strong inflammatory response are all indicators of high IINS, which is typically caused by lymphopenia, hypoalbuminemia, and elevated hs-CRP (*Hua et al., 2020*). Value-added, invasion, and metastasis of malignant tumors are strongly correlated with the inflammatory response that occurs throughout the human body and is mostly mediated by diet and immunology (*Shankaran et al., 2001*; *Swierczak et al., 2015*). Hypoalbuminemia weakens the immune system at large, which promotes the growth of tumor cells (*Cong et al., 2020*). IINS is also a pretty decent predictor of the prognostic value of breast cancer. Lymphocytes are the basis of the cell-mediated antitumor immune response, which inhibits tumor cell proliferation and metastasis (*Muangto et al., 2022*). Low lymphocyte counts are associated with a poor prognosis in a number of malignancies. Lymphocyte counts reduce immune surveillance against cancer (*Zheng et al., 2020*). Consequently, IINS may be a useful, efficient, and easily accessible clinical prognostic indication for people with breast cancer.

In a comparative ROC analysis of IINS, PLR, and PNI, it was found that all three metrics demonstrated significance both for PFS and OS. In order to further clarify the clinical application value of each metric, a comparison was conducted of sensitivity, specificity, positive predictive value (PPV), and negative predictive value (NPV). The results showed that the PPV and NPV assessed by IINS were 65% and 54% for PFS and 41% and 39% for OS. These PPV and NPV values for IINS were higher than those for PLR and PNI, a result that is potentially clinically important to facilitate personalized care for breast cancer (BC) patients and simplify follow-up. However, the PPV and NPV results of IINS are not ideal but only favorable to PLR and PNI. For this reason, IINS should only be used as a supplementary indicator in future clinical practice. The results of this study should also be viewed with caution.

## LIMITATION

The key advantage of the current study is that we employed a newly developed indicator, IINS, which was based on hs-CRP, LYM, and ALB and may perform better than some of the conventional indicators generated from inflammatory, immunological, and nutritional components (*Song et al., 2021*). However, there are some limitations to our study. First, despite the fact that the IINS threshold value was determined from clinical reference

values, there were variations in sample size and patient selection criteria across studies, which will inevitably bias the results (*Wang et al., 2024*). Additionally, while the system is uncomplicated and pragmatic in its categorization, it operates under the assumption of equal weighting, which may not fully capture the inherent biological complexity of the biomarkers in question. Consequently, the interpretation of results should be approached with a degree of circumspection (*Cheng et al., 2025*; *Zhang et al., 2022*).

Although we built a validation cohort for internal validation, the lack of external validation compromises the robustness and generalizability of the study. Secondly, this study involved a relatively small sample size and included patients from only one center, and the conclusions may be potentially biased. To confirm our findings, multicentre, large-scale prospective studies are needed. Thirdly, the relationship between postoperative IINS changes and the prognosis of these patients has not yet been investigated in our study, which focused on the prognostic significance of preoperative IINS. Longitudinal studies should be performed in the future to verify the prognostic significance of the fact that many patients received multiple treatments for tumor recurrence during the follow-up period, which also influenced the prognosis. Finally, our study included a small amount of clinical data that could be used in the future to characterize the dynamics of patients as their disease progresses by statistically analyzing their comorbidities and other biochemical data, and the results may be more instructive in the future.

## CONCLUSION

According to our findings, preoperative IINS may be a potent prognostic predictor in patients with operable breast cancer. In the case of prognosis in particular, IINS is superior to PLR or PNI. Suggesting that it may give a straightforward way of identifying individuals with bad prognosis and the chance to lead therapy and follow-up efforts to improve their status.

### Funding
This work was supported by the Wuxi Taihu Lake Talent Plan, Support for Leading Talents in Medical and Health Professions (Mading academician, 4532001THMD), Wu Jinping Medical Foundation (320.6750.2024-6-100), and the General Project of Wuxi Health Commission (M202408, M202418). The funders had no role in study design, data collection and analysis, decision to publish, or preparation of the manuscript.

### Grant Disclosures
The following grant information was disclosed by the authors:
Wuxi Taihu Lake Talent Plan.
Support for Leading Talents in Medical and Health Professions: 4532001THMD.
Wu Jinping Medical Foundation: 320.6750.2024-6-100.
General Project of Wuxi Health Commission: M202408, M202418.

## Competing Interests

The authors declare that they have no competing interests.

## Author Contributions

- Yuan Wang conceived and designed the experiments, performed the experiments, analyzed the data, prepared figures and/or tables, authored or reviewed drafts of the article, and approved the final draft.
- Wenxin Gao analyzed the data, prepared figures and/or tables, and approved the final draft.
- Shan Wang analyzed the data, prepared figures and/or tables, and approved the final draft.
- Jiajia Zhang analyzed the data, prepared figures and/or tables, and approved the final draft.
- Jiaru Zhuang analyzed the data, prepared figures and/or tables, and approved the final draft.
- Yibo Wu analyzed the data, prepared figures and/or tables, and approved the final draft.
- Xiaoyan Huang conceived and designed the experiments, authored or reviewed drafts of the article, and approved the final draft.
- Jing He conceived and designed the experiments, authored or reviewed drafts of the article, and approved the final draft.

## Human Ethics

The following information was supplied relating to ethical approvals (*i.e.*, approving body and any reference numbers):

This study was approved by the Medical Ethics Committee of the Affiliated Hospital of Jiangnan University, Wuxi City, Jiangsu Province, China (JNMS01202300207).

## Data Availability

Raw data is available in the Supplemental Files.

## Supplemental Information

Supplemental information for this article can be found online at http://dx.doi.org/10.7717/peerj.19950#supplemental-information.

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
