# Peer review of "Significance of the inflammatory-immune-nutritional (IINS) score on postoperative survival and recurrence in breast cancer patients: a retrospective study"

_PeerJ, doi:10.7717/peerj.19950_

## Round 0.1 · original submission · Major Revisions

Dear authors,
Your manuscript holds promise, however, some aspects need revision and re-confirmation (or calculation), namely in terms of statistical analysis. Please, refer to the reviewers' comments for further details.

·

Basic reporting

This manuscript is written in a good writing structure, and written in good English. Figures are also well presented.

Experimental design

Research question is clearly stated, the research method chosen is appropriate, the statistical analysis technique used is also appropriate. In my opinion, there is no conflict in terms of ethics, as long as the researcher has included the ethical clearance of his research.

Validity of the findings

The results of this study have also been presented completely and clearly, and are easy for readers to understand. The researcher has also conveyed the novelty of his research, in this case the use of this new tool in breast cancer patients. The conclusion has also been written completely, including the limitations of this study.

Additional comments

There are 2 inputs from me regarding the results of this study. The first is that this study did not show a significant relationship between the ER, PR, HER2 variables with OS and PFS from both univariate and multivariate analysis. and this is different from what we currently adhere to. the author needs to clarify this. The second is in the discussion section, it is stated that IINS was a distinct risk factor for clinical outcome in BC patients. The terminology risk factor is considered inappropriate. Clarification is needed from the author.

Reviewer 2 ·

Basic reporting

1.The definitions of Progression Free Survival is not given, nor the standard tests used to assess follow up.
2.The definition of IINS is different from other studies previously published (e.g. Li XY, Yao S, He YT, Ke SQ, Ma YF, Lu P, Nie SF, Wei SZ, Liang XJ, Liu L. Inflammation-Immunity-Nutrition Score: A Novel Prognostic Score for Patients with Resectable Colorectal Cancer. J Inflamm Res. 2021 Sep 10;14:4577-4588. doi: 10.2147/JIR.S322260. PMID: 34531673; PMCID: PMC8439969.), which used different cut-offs for the components. The authors seem to have made up a new modified IINS specific to this study, in which it should be named as such.
3. The reference for PNI seems to be wrong.
4. The English is adequate mostly but occasionally requires improvement
5. In Exclusion criteria, it is metioned: Exclusion criteria: "(1) Patients with a history of a malignant tumour or a combination of other primary tumours". It should be "malignant tumor other than breast cancer" in order to be unambiguous

Experimental design

1. Data about the follow up time (median follow up) seems to be missing.
2. The sampling of the cases is not given (consecutive, random or convenience).
3. It is often not clear what is being measured (e.g. is it IINS high versus low, or IINS as a continuous variable). Similar case is for other variables, PLR and PNI. Therefore, it is difficult to reproduce the results by the authors
4. There is doubt about the inclusion criteria: "Patients with histologically confirmed nonmetastatic invasive female breast cancer", yet there were 49 stage III tumors. How is it possible without no lymph node metastasis?

Validity of the findings

1. The survival results by Cox regression, the results do not match that which the reviewer double checked from the raw data. For example, for IINS, this reviewer is getting a hazard ratio of 6.646 for IINS score high vs low , at a cut off or 2, or a hazard ratio of 1.76733, when the IINS score variable is taken as a continuous value. This does not match the results reported.
2. Area under curve with sensitivity and specificity is used to assess survival as a supplementary analysis. This is wrong since the estimates will be biased. If an index is to be used, a Concordance index for survival (as found in coxph function of "survival" package or R or similar index may be used).
3. The reviewer could not understand : "We were able to identify breast cancer patients for whom the IINS, combined with the area under the ROC curve and multivariate Cox regression analysis, was a reliable predictor of overall survival."

Additional comments

The inclusion criteria, Patient information and selection, details about follow up and definitions of the observed variables (e.g. IINS) and outcome measures should be explained in detail.
The statistical calculations should be properly explained. Please check if there is any statistical error, and please give the code for analysis if possible.
It is strongly suggested to use more appropriate measures for survival in place of the ordinary AUC.
The language may be written more clearly.
An explanation of the variables in the submitted raw data is advised.

Reviewer 3 ·

Basic reporting

The writing in this manuscript is clear and unambiguous, with sufficient background and literature references provided. However, some improvements could be made to enhance the clarity of certain tables. Overall, the study is self-contained and presents relevant results aligned with its hypotheses.

Experimental design

Not applicable

Validity of the findings

All underlying data have been provided, and the methods used are robust and statistically sound. The conclusions are clearly stated, directly linked to the original research question, and appropriately limited to the supporting results.

Additional comments

This study investigates the prognostic value of the Inflammation-Immune-Nutritional Score (IINS) in predicting postoperative OS and PFS in patients with breast cancer. A key strength of this research is its comparative analysis and robustness of statistical method. Overall, this study is well-structured, methodologically sound, and clinically relevant. There are several aspects I would like to highlight to enhance the clarity and coherence of this manuscript.

Material and Methods Section

1. In the research design subsection (lines 94–96), the sentence "The stage of each patient's cancer was determined according to the eighth edition of the American Joint Committee on Cancer (AJCC) staging manual" could be considered for removal, as it is already explained in the data collection subsection.

2. In the variable definition section, you describe the calculation of the IINS score based on X-tile software analysis for determining the optimal cutoff associated with OS, which was then further transformed into a scoring system. However, there are several issues related to this approach.

Transforming the cutoff ranges for lymphocyte count, albumin, and PLR into scores of 0, 1, and 2, although also used in previous IINS studies [1–6], is inherently arbitrary. This method assumes equal weighting of each score category, implying that the risk increment from score 0 to 1 is the same as that from 1 to 2. Such an assumption may not accurately reflect the true biological relationship between these biomarkers and the outcome. In reality, the clinical significance of moving from one category to another may not be linear or equal. Furthermore, each category encompasses a range of values, yet individuals with widely different biomarker levels may receive the same score. For instance, a patient with an albumin level of 34.7 g/L and another with 9.2 g/dL would both be scored as 2, despite their markedly different clinical profiles.

I suggest considering the use of weighted regression scores derived from multivariate regression analysis for each biomarker and its association with the outcome. Alternatively, if authors decided to maintain the current scoring system, it would be important to acknowledge this limitation, namely that while the system is simple and practical for classification, it assumes equal weighting and may not capture the true biological complexity of the biomarkers involved. However, if a weighted score generation analysis had been conducted previously but has not yet been reported, the details can be added in the revised manuscript.

Reference:
1. Cheng, Hao, et al. "IINS Vs CALLY Index: A Battle of Prognostic Value in NSCLC Patients Following Surgery." Journal of Inflammation Research (2025): 493-503.
2. Zhang, Zilong, et al. "Prognostic value of inflammation‐immunity‐nutrition score in patients with hepatocellular carcinoma treated with anti‐PD‐1 therapy." Journal of Clinical Laboratory Analysis 36.5 (2022): e24336.
3. Li, Xin-Ying, et al. "Inflammation-immunity-nutrition score: a novel prognostic score for patients with resectable colorectal cancer." Journal of Inflammation Research (2021): 4577-4588.

Results and Discussion Section

1. The information presented in Tables 3 and 4 is not self-explanatory. Consider specifying the reference category or the investigated category for each categorical independent variable. For example in Table 3, instead of writing 'IINS,' you could write 'IINS > 2,' or instead of 'PNI,' write 'PNI > 45.15,' so that readers do not need to refer back to Supplementary Table 1 to understand the cutoff points. This clarification should also be applied to other variables in the table, such as age, TNM stage, tumor size, and node positivity.

2. The same clarification also applies to the reporting of analysis results in lines 172–183. Adding the specific cutoff or category for each variable will enable readers to fully understand which category within each independent variable is associated with increased or decreased OS/PFS.

3. While higher AUC values indicate better discriminatory ability for a particular outcome, the comparison of AUCs across different parameters needs to be tested for statistical significance to determine whether observed differences are meaningful. I suggest conducting a ROC comparison analysis to formally assess whether the difference in AUC values between PLR, PNI and IINS is statistically significant. Additionally, beyond AUC, other performance metrics such as sensitivity, specificity, PPV, and NPV should be reported to comprehensively describe the clinical utility of each biomarker. It would be useful to discuss which performance metric is prioritized in a prognostic setting (e.g. maximizing sensitivity to ensure high-risk patients are identified or maximizing specificity to reduce false positives) and in which clinical scenarios each biomarker might be most useful.

4. Please standardize the number of decimal places reported throughout the results. For example, some tables present p-values with two decimal places, while others use three.

---

## Round 0.2 · Major Revisions

Dear authors,

Thank you for your re-submission.

While some issues may have already been addressed in your revised manuscript, several aspects remain unclear or questionable. We strongly encourage you to ensure transparency in your methods and provide sufficient detail to allow replication of your analyses.

See the reviewers comments. The Cox regression results for your primary variable of interest (IINS) appear to be incorrect. Recalculate the hazard ratios for IINS. If your results differ from the reviewer’s findings, please provide your exact statistical code and methodology so we can trace the discrepancy. Consider including IINS as a continuous variable (without cutoffs), as advised, to assess the robustness of your findings. The definition of progression-free survival (PFS) lacks clarity. Please specify how "progression" was defined and assessed in your study.

For example:
What imaging modalities were used (CT, MRI, etc.)?
Did progression include increase in lymph node involvement, distant metastasis, or other clinical criteria?
Was it based on standardized criteria or clinical judgment alone?
The study-specific cutoff used for IINS optimization may result in overestimated predictive performance. This limitation should be clearly acknowledged in your discussion. Additionally, please consider and report the predictive performance of IINS as a continuous variable, as suggested.

Please revise the manuscript accordingly and resubmit along with a detailed response to each reviewer comment, including any supporting materials such as code or output where relevant.

We look forward to receiving your revised submission.

·

Basic reporting

This manuscript writes about the role of IINS variables on PFS and OS in breast cancer patients, which research was conducted retrospectively. and then compares the IINS variables with similar variables that have been used previously.
Generally written systematically, following appropriate methods, and in good English.

Experimental design

This research is well-conceived, and uses acceptable research methodology. Statistical analysis is done with acceptable tools for drawing conclusions.

Validity of the findings

The findings of this study are scientifically acceptable, because they were conducted with acceptable research methodology.
The first research objective to see the relationship between IINS and PFS and OS has also been carried out with results that we can see together.
Likewise, the second objective to compare IINS with other tools has also shown its results.
So we can accept both results, although there are still questions that arise regarding other things.

Additional comments

The crucial thing that I question is the exclusion of molecular subtype variables in this study. Even though we all know that these variables greatly affect PFS and OS.

Reviewer 2 ·

Basic reporting

1. The definition of PFS has been given, but is limited. What did the authors mean by progression? How was progression assessed. Does this include CT scan or MRI examination? Increase in number of lymph nodes? By what modality? Or is it on clinical findings alone?
2. For IINS score, the study used a cut-off specific to this particular study. This optimization will give optimistic results. This has to be pointed out. In mitigation, it is advised to calculate the Hazard rate of IINS as a continuous variable without any cut offs. This will still show a strong predictive result. This may be pointed out as evidence of inherent strength of IINS.

Experimental design

Ok

Validity of the findings

The results of the Cox regression seem to be wrong. It is laudable that the authors have supplied the raw data to find these mistakes. This also means that mistakes will be caught. The Hazard ratio for IINS, as defined by the authors is markedly diiferent from that calculated by the reviewer using Stata and R. The authors are advised to re-calculate. If they find no errors, please provide the underlying code so that the reviewers may find where they went wrong.

Reviewer 3 ·

Basic reporting

Clear and with good English used throughout the manuscript.

Experimental design

Research question is clear, methods are well described.

Validity of the findings

The conclusions are clear and linked to the research question.

Additional comments

-

---

## Round 0.3 · Major Revisions

Dear authors,
thank your for your re-submission. I think you forgot to address a few aspects in your rebuttal/revision:

1) the exclusion of the molecular subtype variable in this study. Even though we all know that this variable greatly influences PFS and OS.

2) responses that should be improved and incorporated into the manuscript: - concept of progression as per in your work and careful description of its assessment; you acknowledged the a study-specific cut-off for the IINS variable but i failed to understand the "why" behind this; also how could SPSS give results so different from R? (maybe something to address in the manuscript itself too).

So please, revise your previous addressing of the last review round comments.

And then I can request the input of the reviewers again. Many thanks.

---

## Round 0.4 · Minor Revisions

Please, address the reviewer's concerns and suggestions. Many thanks in advance.

**PeerJ Staff Note**: Please ensure that all review, editorial, and staff comments are addressed in a response letter and that any edits or clarifications mentioned in the letter are also inserted into the revised manuscript where appropriate.

**PeerJ Staff Note**: It is PeerJ policy that additional references suggested during the peer-review process should only be included if the authors agree that they are relevant and useful.

Reviewer 2 ·

Basic reporting

Acceptable

Experimental design

Acceptable

Validity of the findings

There still seems to be some problems with the result.
1. The raw IINS score (column IINS in raw data file) seems to have been used for univariate analysis, but the column "IINS score" seems to have been used for multivariate analysis. Both univariate and multivariate analysis should be the same
2. The "Lymph node positivity" used in calculation of PFS score is NOT Lymph node positivity per se, but number of Lymph nodes found positive. Strongly suggest to use lymph node positivity as per Lymph node positive versus negative in the calculations both for multivariate as well as univariate analysis.
3. In the multivariate analysis for OS, the results for TNM and Size are not matching. This seems to be a copying mistake from results in the manuscript since the results in the rebuttal letter matches that done by me.
4. The tables should mention the nature of the variables , i.e. is the IINS score being evaluated or the raw IINS. Also whether it is number of positive lymph nodes or Lymph node positivity, whether it is size measured in cm , Age in Years etc

The full analysis on Raw data with the commands is given in the pdf attachment.


I still have concerns about usage of Area under AUC curve. In typical ROC analysis, outcomes are binary (event vs. no event), observed fully for everyone.

In survival data, not all individuals experience the event (due to right censoring) — so the true outcome is unknown for some.If you treat censored observations as event-free, you underestimate the true discrimination.
Proper alternatives for time-to-event data include
1. Time-dependent ROC curves
2. Uno's C-index (an extension of AUC)

Harrel's C index (given as Concordance in the results above) is partially accurate, though it only partially accounts for censoring
It ignores pairs where the ordering is unclear due to censoring. I am unsure if these methods are implemented in SPSS, so I have included the Harrel's concordance in the results. There is no necessity for graphs.

Reference: Park SY, Park JE, Kim H, Park SH. Review of Statistical Methods for Evaluating the Performance of Survival or Other Time-to-Event Prediction Models (from Conventional to Deep Learning Approaches). Korean J Radiol. 2021 Oct;22(10):1697-1707. https://doi.org/10.3348/kjr.2021.0223

Annotated reviews are not available for download in order to protect the identity of reviewers who chose to remain anonymous.

---

## Round 0.5 · accepted · Accept

Dear authors,
Thank you for your patience and work. I am now accepting your work for publication. Thank you for submitting to PeerJ.

Reviewer 2 ·

Basic reporting

Satisfactory

Experimental design

Satisfactory

Validity of the findings

Satisfactory